# Drug-Targeted Genomes: Mutability of Ion Channels and GPCRs

**DOI:** 10.3390/biomedicines10030594

**Published:** 2022-03-03

**Authors:** Regan Raines, Ian McKnight, Hunter White, Kaitlyn Legg, Chan Lee, Wei Li, Peter H. U. Lee, Joon W. Shim

**Affiliations:** 1Department of Biomedical Engineering, College of Engineering and Computer Sciences, Marshall University, Huntington, WV 25755, USA; raines83@marshall.edu (R.R.); mcknight31@marshall.edu (I.M.); white869@marshall.edu (H.W.); legg144@marshall.edu (K.L.); 2Indiana University Health Arnett Hospital, Lafayette, IN 47905, USA; clee9@iuhealth.org; 3Department of Biomedical Sciences, Joan C. Edwards School of Medicine, Marshall University, Huntington, WV 25755, USA; liwe@marshall.edu; 4Department of Pathology and Laboratory Medicine, Brown University, Providence, RI 02912, USA; peter_lee@brown.edu; 5Department of Cardiothoracic Surgery, Southcoast Health, Fall River, MA 02720, USA

**Keywords:** target therapy, ion channels, GPCRs, telomere, mutation, cardiovascular pharmacology

## Abstract

Mutations of ion channels and G-protein-coupled receptors (GPCRs) are not uncommon and can lead to cardiovascular diseases. Given previously reported multiple factors associated with high mutation rates, we sorted the relative mutability of multiple human genes by (i) proximity to telomeres and/or (ii) high adenine and thymine (A+T) content. We extracted genomic information using the genome data viewer and examined the mutability of 118 ion channel and 143 GPCR genes based on their association with factors (i) and (ii). We then assessed these two factors with 31 genes encoding ion channels or GPCRs that are targeted by the United States Food and Drug Administration (FDA)-approved drugs. Out of the 118 ion channel genes studied, 80 met either factor (i) or (ii), resulting in a 68% match. In contrast, a 78% match was found for the 143 GPCR genes. We also found that the GPCR genes (*n* = 20) targeted by FDA-approved drugs have a relatively lower mutability than those genes encoding ion channels (*n* = 11), where targeted genes encoding GPCRs were shorter in length. The result of this study suggests that the use of matching rate analysis on factor-druggable genome is feasible to systematically compare the relative mutability of GPCRs and ion channels. The analysis on chromosomes by two factors identified a unique characteristic of GPCRs, which have a significant relationship between their nucleotide sizes and proximity to telomeres, unlike most genetic loci susceptible to human diseases.

## 1. Introduction

Ion channels are ubiquitous membrane proteins critical to almost all physiological processes, such as muscle contraction, neural transmission, and cardiac pace-maker function [1,2,3]. Ion channels are macromolecules with a relatively long size that are composed of subunits spanning the cell membrane. These channels are classified by the ion they allow the passage of—sodium (Na^+^), potassium (K^+^), calcium (Ca^2+^), or chloride (Cl^−^). Ion channels function as a gate via opening or closing by extracellular ligands [4], transmembrane voltage changes [5], or intracellular second messengers [6,7]. Mutations in ion channels are either causative or contributory to the pathogenesis of numerous disorders, such as cystic fibrosis [8,9], long-QT syndrome of the heart [10,11,12,13,14], heritable hypertension (e.g., Liddle’s syndrome) [15,16,17,18], hyperinsulinemia and hypoglycemia of infancy [19,20,21], hereditary nephrolithiasis (e.g., Dent’s disease) [22,23,24], and certain hereditary myopathies [25,26,27].

Ion channel proteins have been extensively investigated as drug targets, but it took much longer than expected for the outcomes of this research to be widely adopted clinically. Amlodipine besylate (brand name, Norvasc; Pfizer), which acts by blocking voltage-gated Ca^2+^ channels [28,29,30,31,32], was patented in 1982, and it has since become one of the bestselling anti-hypertensive medications. In 2019, amlodipine was the fifth most prescribed medication in the United States (US) [33]. As such, it has been perceived by drug developers that modulating ion channels is harder than targeting enzymes, kinases, and G-protein-coupled receptors (GPCRs). When designing a new drug, some would rather avoid interactions with ion channel proteins, particularly due to cardiovascular safety concerns. Although several attempts have been made to target ion channels, such as K^+^ channel blockers for autoimmune disease [34,35] and antiseizure medications [36,37,38,39,40], and antiarrhythmic treatments [41,42,43,44,45,46], targeting ion channels has resulted in a limited commercialization, as evidenced in the development of Ca^2+^ channel blockers for hypertension [47,48]. Nevertheless, a pathway for therapeutic success [49] and significant progress have been achieved with drugs that target Cl^−^ channels, such as cystic fibrosis transmembrane conductance regulator (CFTR) [50]. Moreover, Lidocaine, one of the most commonly prescribed medications in the US, is known to successfully target Na^+^ channels [51,52]. To a large extent, G-protein-coupled receptors (GPCRs) are vigorously studied as drug targets in the modern age [53], since they regulate diverse physiological processes by functioning as transmembrane transducers, which carry signals from extracellular ligands to effectors within the cell [54]. GPCRs can be linked to ion channels through their ligand, G protein, which is mediated by the direct physical interactions between G protein subunits and ion channels, including numerous voltage-dependent Ca^2+^ channels, as well as G-protein-activated K^+^ channels [55,56,57,58]. The majority of transmembrane signaling, which is accomplished by neurotransmitters and hormones, is largely dependent on the activation of G proteins by GPCRs [59]. In fact, nearly all druggable targets belong to one of the five primary families of proteins: GPCRs, ion channels, kinases, nuclear hormone receptors, or proteases [60]. Out of these five, GPCRs have been studied most intensively. In fact, nearly 30% of the global market share of therapeutic agents are associated with GPCR drugs [60,61]. GPCRs are the primary means by which cells can detect stimuli in the environment, as well as communicate with one another [62]. To this end, more than 60% of human hormones and roughly 30% of clinical drugs are able to activate approximately 350 GPCRs [62].

While the preference for GPCRs over ion channels as drug targets differs depending on the priority of manufacturers, the risk of targeting these proteins lies in unexpected mutations. Genetic mutation in the human genome is able to cause different responses to medications and is an increasing burden on public health. While 108 GPCRs are currently targeted by 34% (475) of Food and Drug Administration (FDA)-approved drugs, which account for more than 180 billion US dollars in annual sales globally, the likelihood and prevalence of mutations among drug-targeted GPCRs remain to be confirmed [61]. One previous report by the 1000 Genomes Project showed that an individual harbors an average of 68 missense mutations in the coding region of 33% of the GPCR drug targets [63]. Of these, eight mutations have been associated with clinically altered drug responses. Although ion channels are less preferred as drug targets in comparison to GPCRs, mutations of the host (human) gene leading to drug resistance have been previously reported in the North Indian population [64]. These data suggest a differential role of genetic polymorphisms of sodium voltage-gated channel alpha subunit 1 (SCN1A) and subunit 2 (SCN2A) in terms of both susceptibility to epilepsy, as well as drug response [64]. Three specific gene characteristics have been previously correlated with high rates of mutation in human chromosomes: (1) recombination rate [65], (2) proximity of a gene to its telomere, and (3) high adenine/thymine (A+T) content of a gene [66,67]. Of these three factors, we have shown previously that merely two of these factors—proximity to a telomere [68] and A+T content—are sufficient to predict the susceptibility of the genes to mutations and lead to monogenic and/or polygenic diseases [69,70]. To gain a better understanding through determination of the disease phenotype, the National Institute of Health (NIH) has released a list of 390 novel druggable genes under the title of ‘Commercializing Understudied Proteins from the Illuminating the Druggable Genome’ project (PA-19-034). Out of the 390 genes listed, 118 of the druggable candidates were classified as ion channels, while 143 of the genes were classified as GPCRs. Because genes encoding proteins that are more prone to mutations make less attractive drug targets, having a way to predict the mutability of these >260 ion channel and GPCR genes would help to prioritize which druggable targets to pursue for commercialization. 

In this report, we analyze how many of the genes that encode the 118 druggable ion channels and 143 GPCR genes identified by the NIH are able to match with one, both, or neither of the two previously identified predictive factors for genetic mutations: (i) proximity to telomeres and (ii) high A+T content. This predictive mechanism allows us to prioritize the 261 druggable proteins based on their relative mutability, as predicted by these aforementioned two factors. We then apply the same two factors to the 31 genes encoding ion channels and GPCRs targeted by currently FDA-approved drugs, providing the matching rate of these two factors with their respective drug targets, which include treatments for a wide range of human diseases in the cardiovascular, neurological, and other major organ systems. We demonstrate that this factor-druggable gene matching rate analysis is a useful tool to help systematically determine the relative mutability of these 261 novel druggable genes encoding ion channels and GPCRs, and thus inform the prioritization of drug development.

## 2. Materials and Methods

### 2.1. Database, Literature, and Open-Access Software

The publicly open NIH program announcement (PA-19-034) detailing the classification and identification of 390 understudied druggable genomes was utilized to acquire the list of 261 candidate genes encoding GPCRs and ion channels. The associated literature survey was performed until 9 December 2021, with emphasis on published title words with (1) ion channels since 2017 [4,5,9,10,11,12,13,14,16,26,27,30,33,39,40,41,42,44,46,47,48,53,54,59,61,69,70,71,72,73,74,75,76,77,78,79,80], (2) ion channel blockers since 2017 [41,81,82,83,84,85,86,87,88,89,90,91,92,93,94,95], (3) GPCR drugs since 2017 [53,60,61,96,97,98,99,100,101,102,103,104,105,106,107,108,109,110,111,112,113,114,115,116,117,118,119,120,121,122,123,124,125,126,127,128,129,130,131,132,133], (4) hypertension and GPCR [134,135,136,137,138,139,140,141,142,143], (5) AD and GPCR [144,145,146,147,148], and (6) Down syndrome [149,150,151,152,153,154,155,156,157,158,159,160,161,162,163,164,165,166,167,168,169,170,171,172,173,174,175,176,177,178,179,180,181,182,183]. This literature survey was systematically conducted following the method that has been used previously [69].

In order to measure the distance between a designated gene of interest and its telomere, we utilized the NIH Genome Data Viewer (https://www.ncbi.nlm.nih.gov/genome/gdv/) accessed on 1 February 2022. To calculate the A+T content present in a designated gene of interest, a GC content calculator (https://www.biologicscorp.com/tools/GCContent/#.XvctCi-z2uV) accessed on 1 February 2022 was used. This resulted in the acquisition of the compositions of adenine and thymine along with the full-length sizes of the nucleotide [69,70] for each gene.

### 2.2. Approximation of Proximity to a Telomere

We have followed a previously established method in approximating the proximity of a gene to its telomere [70]. Therefore, in this study, we have determined the position of each gene encoding an ion channel or GPCR and its distal end locus of each arm (telomere) according to the following principle:

If the recombination frequency is less than (≤) 50 centimorgan (cM), genes are linked and are therefore proximal to one another: (1) If the recombination frequency is higher than 50 cM, genes are not linked and are therefore distal to one another: (2) (where 1 cM ≅ 1 million base pair (Mbp) [184]).

### 2.3. Data Plot and Statistical Methods

Relevant heatmap plots, bar graphs, as well as box and violin plots of the data obtained during analysis with the genome data viewer were generated via the use of Prism. Similarly, all statistical analyses were performed using Prism (version 9.3.0, GraphPad Software Inc., San Diego, CA, USA) as well. A Shapiro–Wilk normality test (α < 0.05) was used to confirm that the data were normally distributed. Unless stated differently, all comparisons of two groups at a time were conducted through the use of two-sided unpaired t-tests. In order to compare more than two groups at a time, after conducting one-way analysis of variance (ANOVA), Tukey’s multiple comparisons test was used. For all statistical analyses, differences were considered significant at *p* < 0.05. *p*-values are denoted in the figures/legends as * *p* < 0.05, ** *p* < 0.01, and *** *p* < 0.005.

## 3. Results

### 3.1. Outlines of Two Factors in Genes Encoding 261 Novel Druggable Genomes

We sought to examine the relative mutability of the druggable ion channel and GPCR genes belonging to the 361 understudied druggable genomes identified by the National Institute of Health (NIH PA-19-034). The heatmap plots of the 118 ion channels and 143 GPCRs were created to demonstrate (1) relative distribution of the genes across all the chromosomes (1 to 23, in which 23 corresponds to the chromosome X), (2) the distance of a human gene from its telomere in million bases (Mb; 0 to 200), and (3) A+T composition of the gene (%). Specifically, the intensity of the overall tone in color is relatively weak in the chromosomal distribution of the 118 ion channels from chromosome 1 to 22, and X (assigned as 23; Figure 1a, the left panel) being comparatively strong in the distribution of proximity to telomeres (Figure 1a, the middle panel), and at the intermediate range in A+T content (Figure 1a, the right panel) (Appendix A). 

As these data are organized in scatter plots, factor (i) or the location of the genes is noticeably skewed to being closer to their telomeres while still displaying a wide range, whereas factor (ii) forms a distinct cluster near 50% (Figure 1b). Roughly 62% of the 118 genes encoding ion channels satisfied factor (i) located < 50 Mb from the telomere, whereas only ~15% of the 118 ion channel genes met factor (ii) with an A+T content greater than 59%, which is the human genome-wide average [67]. For the 143 genes encoding GPCRs, the range of distances to telomeres was narrower (0 to 150 Mb; Figure 1c) than for ion channels (up to 250 Mb). The number of genes encoding druggable GPCRs that met F(i) (gene proximity < 50 Mb from its telomeres) was greater than for ion channels at ~71% vs. 62%. Consistent with the relative mutability estimated by proximity to telomeres, ~29% of the 143 genes encoding GPCRs satisfied F(ii) at >59% (Figure 1d), whereas a smaller portion or ~15% met F(ii) for ion channels.

### 3.2. Prioritization of Druggable Ion Channels per the Relative Mutability

Next, we were able to identify the matching rate between druggable ion channel genes and the two aforementioned factors. We determined that roughly 68% of the ion channel genes were either sufficiently proximal to telomeres or contained high A+T content (*n* = 80/118). On the other hand, approximately 32% of the genes (38 of 118) encoding druggable ion channels satisfied neither factor. Moreover, fewer than 9% of the genes (10 of 118) met the conditions of both factors. As in the previous reports [69,70], the groups of ’Both (meeting both factors)’ and ‘None (meeting neither factor)’ were particularly noteworthy, since they indicated that a gene was a relatively more or less mutable target, respectively, thereby potentially informing the commercialization potential of drugs targeting ion channels associated with these genes (Figure 2a).

When examining the correlation between the molecular size of a particular gene and each of the two factors, the Pearson coefficient (r = −0.008) indicated no significant correlation (*p* = 0.93) between the full-length sizes of the genes and the proximity of each gene to its nearest telomere in druggable ion channels (Figure 2b). However, as is consistent with the prior report [70], there was a significant correlation between the full-length size of a gene and its A+T content (r = 0.32; *p* = 0.0003) (Figure 2c).

Next, we examined each of the specific genes that matched with ‘both’ or ‘none’ of the two factors, then prioritized the top five ion channels based on their relative mutability. This allowed us to produce the names of five candidate genes (*GABRG1, GLRA3, TMEM38B, PKD2L2,* and *GLRB*), which showed relatively higher mutability corresponding to the ‘Both’ group (orange arrow). Alternatively, five other candidate genes (*CLCN6, CLCNKA, CATSPER4, BEST4*, and *CHRNA9*) were sorted as the ion channel genes with relatively lower mutability corresponding to the ‘None’ group (blue arrow; Figure 2d).

We then analyzed each gene alongside F(i) and F(ii) with respect to the length of each nucleotide. Following previous analyses [69,70], we grouped all analyzed genes into three categories of length: 1–3000 bases (*n* = 47), 3001–6000 bases (*n* = 51), and 6001–17,000 bases (*n* = 20). Through statistical analysis, we determined that there was no significant difference in the proximity of a gene to its telomere with respect to the gene’s full-length size (*p* = 0.77). However, after conducting a one-way analysis of variance (ANOVA) and pair-wise comparisons using post hoc Tukey’s test, it became clear that there was a significant difference (*p* = 0.0002) in the A+T content of a gene with respect to the full-length size (bases) of the gene between the longest and shortest sub-groups. Furthermore, we also determined that there was a significant difference (*p* = 0.0096) between the intermediate and longest genes as well (Figure 2e,f).

Our discovery of this subset of ion channels with a relatively low predicted mutability rate implies that commercialized ion channel drugs could fit into any of these three categories meeting (1) one of the two factors or (2) both or (3) none.

### 3.3. Prioritization of Druggable GPCRs per the Relative Mutability

Moreover, we also compared the matching rate between these two aforementioned factors and 143 druggable GPCRs (Appendix A). Of these GPCRs, 78% of 143 genes met either proximity to telomeres or high A+T content (*n* = 112/143). Notably, more than 20% of the genes (30 of 143) that encode GPCRs satisfied both F(i) and F(ii). On the other hand, 22% of these genes (31 of 143) met neither factor (Figure 3a). To examine the correlation between molecular size and each of the factors, the Pearson coefficient (r = 0.12) was determined. This analysis indicated the existence of a significant correlation (*p* = 0.035) between the full-length sizes of the 143 genes encoding druggable GPCRs and their proximity to telomeres. Unlike previous reports, this significant correlation between molecular size and telomere proximity has never been detected in any genomic analysis [69,70]. However, no significant difference between the full-length size of the GPCR-encoding genes and their associated A+T content was detected (r = 0.019; *p* = 0.8245) (Figure 3b,c).

Next, we assessed each of the specific genes that matched with ‘both’ or ‘none’ of the two factors and prioritized the top five GPCRs of each category based on their relative mutabilities. This strategy enabled us to sort out five candidate genes (*GPR82, GPR34, SUCNR1, TAS2R10*, and *TAS2R13*), which showed a relatively higher mutability, corresponding to the ‘Both’ group (denoted with orange arrow). Conversely, five other candidate genes (*GPR45, FZD10, ADGRD2, GPR156*, and *GPR39*) were selected as the GPCR genes with a relatively lower mutability, which corresponded to the ‘None’ group (denoted with blue arrow; Figure 3d).

We then organized both factors with respect to the length of individual nucleotides. Per the previous analysis [69,70], we grouped all of the examined genes into three categories: 1–3000 bases (*n* = 88), 3001–6000 bases (*n* = 46), and 6001–17,000 bases (*n* = 8), respectively. Statistical analysis conducted on these groups suggested that there was a significant difference in the proximity of a gene to its telomere between the shortest and intermediate size genes (*p* = 0.027). After a one-way ANOVA, pair-wise comparisons using post hoc Tukey’s test revealed no significant difference in the A+T content of a gene with respect to its full-length size (bases) (Figure 3e,f).

### 3.4. Mutability of Ion Channels and GPCRs Targeted by the Commercialized Drugs

To glean a more comprehensive understanding of the genomic characteristics that are associated with mutations in novel druggable genomes in terms of known commercialized drugs, we examined the mechanisms of action of the drugs on the market, which target either ion channels or GPCRs. Primarily, the portion of the survey on commercialized drugs targeting ion channels resulted in eleven pharmacological agents, from those targeting L-type CaV (amlodipine) to those targeting *TRAAK-1* (Riluzole) (Table 1). With the exception of 2 drugs, 9 out of 11 drugs targeting ion channels satisfied either F(i) or F(ii). Specifically, for amlodipine, cana1c demonstrated a sufficient proximity to its telomere at 1.9 Mb. For pregabalin, *CACNA2D1* satisfied F(ii) or high A+T content at 64%. For sotalol, *KCNH2* satisfied proximity to its telomere at 8 Mb. For flecainide, *SCN5A* met F(i) at 38 Mb. For ziconotide, *CACNA1B* met proximity to its telomere at 0.1 Mb. For varenicline, *CHRNA4* satisfied proximity to its telomere at 1 Mb. Similarly, for retigabine, *KCNQ2* met F(i) at 1 Mb. For VU0456810, *GIRK2* met both proximity to its telomere at 9 Mb and high A+T content at 60%. In the case of riluzole, *KCNK4* met neither proximity to its telomere nor high A+T content. Similarly, for diazepam, *GABRB3* satisfied neither F(i) nor F(ii). Among these 11 drugs targeted by ion channels, 82% (9 out of 11) satisfied either F(i) or F(ii) (Table 1).

The measurements of distances to telomeres and nucleotide compositions on genes encoding GPCRs targeted by the commercialized drugs demonstrated that fifteen different genes—encoding *PTGFR, OPRD1, HTR1A, GLP1R, PTGIR, SMO, PTH1R, P2RY12, NR1L2, ADRB2, GLP1R, MTNR1A, OPRM1, ADRB1*, and *HTR1A*—satisfied either proximity to telomeres at < 50 Mb or high A+T content at > 59%. On the other hand, five additional genes encoding GPCRs, *S1PR1, TACR1, HCRT2, CASR*, and *DRD3*, met neither of the two factors (Table 2).

### 3.5. Thirty-One Genes Encoding Ion Channels and GPCRs Targeted by Approved Drugs

We next assessed the two factor characteristics of genes encoding ion channels and GPCRs targeted by 31 commercialized drugs. The heatmap plots of 11 genes encoding ion channels demonstrate a similar spectrum to the 118 druggable ion channels (Figure 1), as assessed on the range of values in chromosome number (1 to 23, in which 23 corresponds to the chromosome X), proximity to telomeres (Mb; 0 to 100), and A+T content (%; between 35 and 70). Specifically, the intensity of the overall tone in color is relatively weak in the chromosomal distribution of 11 ion channels from chromosome 1 to 22, and X (assigned as 23; Figure 4a, the left panel), strong in the distribution of proximity to telomeres (Figure 4a, the middle panel), and at the intermediate range in A+T content (Figure 4a, the right panel). As these data are organized in scatter plots, F(i) shows a widespread distribution over 0 to 100 Mb in proximity to telomeres, while F(ii) forms a narrower distribution near 50% (Figure 4b). Approximately 64% of 11 genes encoding ion channels satisfied F(i) at <50 Mb, while ~36% of 11 ion channel genes met F(ii) at >59%. A similar spectrum in color is found in proximity to telomeres of 20 genes encoding GPCRs (0 to 100 Mb; Figure 4c). As compared to ion channels targeted by approved drugs (Figure 4a), a greater number of genes encoding GPCRs satisfied F(i) at 60% at <50 Mb, while 30% of 20 genes encoding GPCRs met F(ii) at >59% (Figure 4d). To this end, we provide the logical flow of work conducted in the present analyses (Figure 5).

## 4. Discussion

To better understand the mutability of druggable genomes, we applied two factors associated with high mutation rates to the genes encoding ion channels and GPCRs. The novelty of this work lies in our finding that the two factors identify a unique feature of GPCRs, having a significant relationship between their nucleotide sizes and proximity to telomeres unlike the majority of genetic loci susceptible to human diseases [69,70]. In the human genome, it has been reported that more than 400 genes encoding ion channels mediate ion fluxes across cell membranes [190]. A significantly smaller percentage (~15%) of currently used drugs target ion channels [49]. In contrast, more than 800 members of GPCRs are known so far, forming the largest family of cell-surface receptors [120]. As expected from this difference in numbers (400 ion channels vs. 800 GPCRs), 34% of all drugs approved by the US FDA act on over 100 GPCRs, and clinical trials have explored >300 new GPCR agents [120].

Using the outlines of druggable genomes in color-coded maps (Figure 1a vs. Figure 1c), we first noticed that ion channels are bigger in size than GPCRs by the upper limit (250 vs. 150 Mb). Given the previous reports [69,70], this suggested to us that druggable ion channels are less likely to meet the factor (i) or proximity to telomeres than druggable GPCRs are. As displayed in Figure 1b,d, we found that relatively small portions of ion channels (62%) and GPCRs (71%) met the factor (i) when compared with the high matching rates reported in >100 genes at ~80 to 91% [69,70]. Twice as many GPCRs (29%) met the factor (ii) or high A+T content than ion channels did (15%). This also suggested that the relationship or correlation between the molecular size and A+T content in ion channels differed from that of GPCRs. 

The matching rate of either of the two factors with druggable ion channels (Figure 2a) was comparable to those analyzed in genetic mutant loci of congenital heart disease (CHD) and thoracic aortic aneurysm (TAA), which showed a matching rate at ~74% (62 of 84 genes, collectively) [69]. Consistent with the previous reports [69,70], there was a typical non-significant relationship in full-length (FL) size of the druggable ion channels with respect to the proximity to telomeres. A typical feature of statistical significance in FL–A+T content was also noted in druggable ion channels (Figure 2b,c). From the short lists of more vs. less mutable genes, *TMEM38B* draws our attention, as it belongs to one of the five most mutable ion channels due primarily to high A+T content (Figure 2d). When 118 druggable ion channel genes were divided into three groups per their molecular size, a typical correlation between FL–proximity, as well as FL–A+T content, was revealed (Figure 2e,f), which is consistent with the result obtained from 108 genes causing congenital disorder of the brain where *TMEM67* mutants were reported [70,74,77]. Overall, 118 genes encoding druggable ion channels were relatively less mutable and longer in molecular size than 143 genes encoding druggable GPCRs (Figure 2a vs. Figure 3a).

Next, the matching rate of one of the two factors with druggable GPCRs (Figure 3a) was reminiscent of the result obtained from Alzheimer’s disease (AD) and monogenic hypertension (MH), which resulted in the matches of either of the two factors and the disease at ~84% (59 of 70 genes susceptible to AD; 66 of 79 genes causative of MH) [69,70]. Indeed, GPCRs become important targets in understanding the molecular mechanisms underlying AD [144,145,146,147,148] and hypertension [134,135,136,137,138,139,140,141,142,143]. Inconsistent with the prior reports [69,70], however, there was an atypical significant relationship in proximity to telomeres with respect to the molecular size (FL) of the druggable GPCRs (Figure 3b,c). Given the human genome-wide average at 59% [67], most mutable GPCRs sorted in this study have demonstrated their A+T content at unusually high compositions of 65–66% (Figure 3d). When 143 druggable GPCRs were divided into three groups per their molecular size, the atypical correlation between FL–proximity was detected (Figure 3e in comparison with Figure 3f), which is a unique feature of the GPCRs that has never been found in any of the human genes associated with monogenic or polygenic disease analyzed previously [69,70].

As we further analyzed the approved drugs targeting ion channels and GPCRs, it became clear that ion channels with much smaller molecular sizes than the 118 listed druggable genomes are commercialized into drugs, as inferred from their proximity ranges being located between 0 to less than 100 Mb (Figure 4a compared to Figure 1a). As assessed by the molecular sizes of ion channels targeted by approved drugs (Table 1), the observation that drugs are targeting molecules with the nucleotide sizes of less than 10,000 bp is confirmed with three exceptions: (1) amlodipine targeting *CACNA1C* with 13,744 bp, (2) lacosamide targeting *SCN1A* showing 13,079 bp, and (3) VV0456810 targeting *KCNJ6* with 19,657 bp. Our data (Figure 1 and Figure 4) indicate that if one aims at designing new drugs, one will follow the trend revealed in these analyses:
-Figure 1: There are >260 candidates proposed (ion channels and GPCRs), but sizes are wider in range.-Figure 4: In reality, drug manufacturers target genes with small(er) sizes in ion channels and GPCRs.

Why? If the target is long in size, can more side effects arise? Specificity will decrease because the longer the sequences, there is a higher chance of (1) having multiple same/similar fraction of sequences and (2) having higher A+T content.

Even though the smaller-size molecules were chosen to be commercialized into drugs (Figure 4b), the matching rates of proximity to telomeres and ion channels are similar between the druggable and the drug-targeted ion channel genes (62% vs. 64%). As far as mutability is concerned, however, much higher matching rate of ion channels and A+T content that we found from drug-targeted genes (36%) as compared to those of druggable ion channels (15%) may be due to the relatively longer molecular sizes of ion channels (Figure 1b vs. Figure 4b) as compared to those of GPCRs (Figure 1d vs. Figure 4d). 

At a 10% margin for the factor (i), the matching rates of either of the two factors and the gene under investigation are similar between the druggable GPCRs (71% and 29%; Figure 1d) and the GPCRs targeted by approved drugs (60% and 30%; Figure 4d). Collectively, these two factors organized in specific numbers support the idea that the same proportions (2 of 11 genes vs. 5 of 20 genes) of the molecules show the least mutability in 11 ion channels and 20 GPCRs targeted by approved drugs (marked in grey in Table 1 and Table 2).

Three genomic factors of recombination rate, proximity to telomeres, and high A+T content [67], in addition to a defect in cell division or chromosome mis-segregation, resulting in aneuploidy [149,158,163,164,165,169,176], were reported to be associated with high mutation rates [67]. Although the utility of matching rate analysis has been demonstrated with the highest predictability at ~91% in 108 genes [69,70], there is a limitation in the applicability of matching rate analysis to birth defects, such as Down syndrome, which is caused by trisomy 21 [149,158,163,164,165,169,176]. 

## 5. Conclusions

Among the 118 examined druggable ion channels, 80 of the human genes that encode ion channels were suitably proximal to their telomeres at <50 Mb or contained high A+T content at >59%, suggesting that 68% of these genes are mutable. On the other hand, 38 of 118 genes that encode druggable ion channels were determined to be relatively less mutable, since they satisfied neither of the two factors. Of the 143 genes encoding druggable GPCRs, 112 of these genes were able to meet either F(i) or F(ii), suggesting that 78% of these GPCR genes have high mutability. Conversely, 31 of the 143 genes encoding druggable GPCRs were determined to be relatively less mutable, thereby meeting neither of the two aforementioned factors. Moreover, eleven ion channels and twenty GPCRs targeted by FDA-approved drugs were suggestive that ion channels have relatively longer molecular sizes than GPCRs, resulting in a higher likelihood of meeting F(ii). Overall, 118 genes encoding druggable ion channels were relatively less mutable and longer in molecular sizes than 143 genes encoding druggable GPCRs. Compared to the approved drugs targeting ion channels and GPCRs, however, ion channels with much smaller molecular sizes than 118 druggable ion channels are targeted on the market and commercialized as drugs. Similarly, relatively smaller-size GPCRs at the range of less than 10,000 bp are chosen to be materialized into GPCR drugs. Both cases of intended selections in shorter-size molecules support the previous findings [69,70] that a shorter nucleotide length of a gene under investigation corresponds to a reduced likelihood to satisfy the factor (ii) or high A+T content, leading to drug targets with relatively less mutability.

Investigators in the field use laboratory animals aiming to test their hypotheses ultimately for translational purposes. It is not uncommon to find that a pharmacological approach that worked in rodents fails to reproduce the expected outcome when tested in human clinical trials. This study provides a guideline of prioritizing a more reliable drug target by relative mutability based on two factors using the genomes of humans. With all conditions, such as safety, toxicity, and efficacy of drugs, comparable, and if there are two or more such drug targets, a less mutable pharmacological target can be determined, augmenting decision making on which drug target will result in a consistent outcome in laboratory animals and humans. This means that the same analysis presented in this study can be applied to diverse animal genomes available at the National Center for Biotechnology Information database.

## Figures and Tables

**Figure 1 biomedicines-10-00594-f001:**
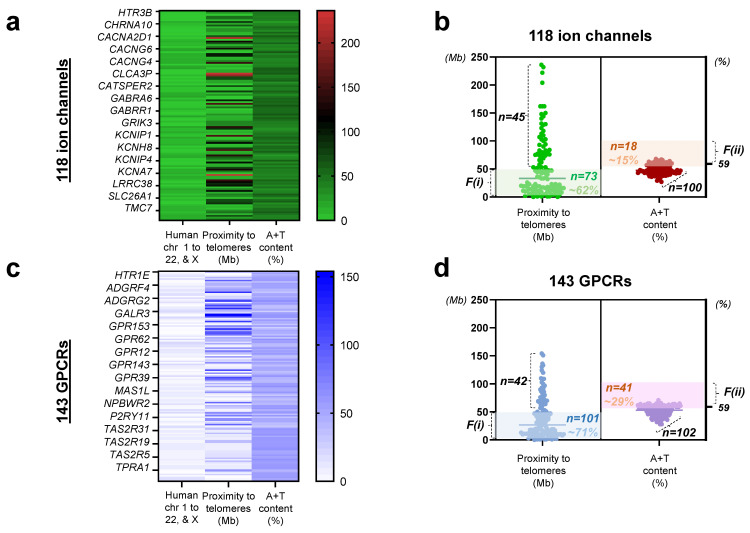
Relative mutability of genes encoding ion channels and GPCRs (**a**) a color-coded map illustrating distributions of 118 genes encoding ion channels with respect to the human chromosome 1 through 22 and X (color code range 1 to 23), proximity to telomeres, and A+T content (range 0 to 100%), respectively. Note that only select genes (not all 118) are marked on the left y axis. The reference proximity to telomeres (range 0 to 250 Mb) with coded colors is shown on the right. (**b**) Scatter plots displaying relative mutability of druggable ion channels by proximity to telomeres (left) and A+T content (**right**). Shaded regions with light green (**left**) and light brown (right) demonstrate genes satisfying either of the two factors. Genes outside the shaded regions (*n* = 45 by F(i); *n* = 100 by F(ii)); relatively less mutable. (**c**) the color-coded map illustrating distributions of 143 genes encoding GPCRs with respect to the human chromosome 1 through 22 and X (color code range 1 to 23), proximity to telomeres (range 0 to 150), and A+T content (range 0 to 100), respectively. Note that select genes (not all 143) are marked on the left y axis. The reference number with coded color is shown on the right, (**d**) scatter plots exhibiting relative mutability of druggable GPCRs by proximity to telomeres (**left**) and A+T content (**right**). Shaded regions with light blue (**left**) and light purple (**right**) show genes satisfying either of the two factors. Genes outside the shades (*n* = 42 by F(i); *n* = 102 by F(ii)) are relatively less mutable.

**Figure 2 biomedicines-10-00594-f002:**
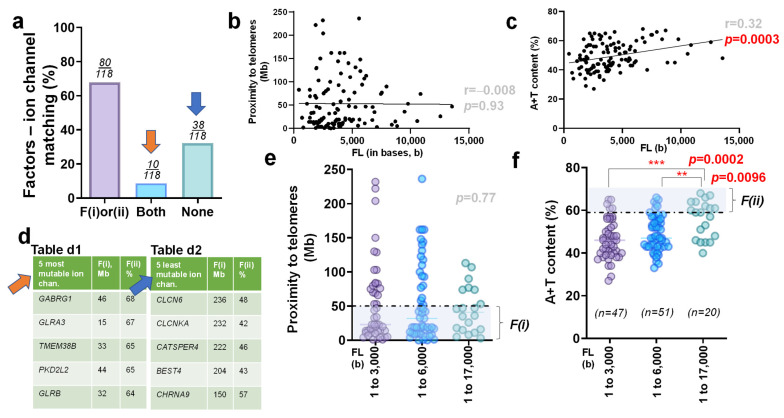
Relative mutability of druggable ion channels by two factors (**a**) Bar graph, which demonstrates the matching rate of genes encoding 118 druggable ion channels with both telomere proximity, F(i), and nucleotide composition, F(ii). Genes are sorted into three separate groups: those that satisfy one of the two factors, (F(i) or (ii)), those meeting both F(i) and F(ii) (Both, denoted with orange arrow), and those satisfying neither F(i) nor F(ii) (None, denoted with blue arrow) (**b**) Scatter plot showcasing the correlation between the full-length size of genes that encode ion channels and their proximity to telomeres. The Pearson correlation coefficient for this relationship is r = −0.008, with a level of statistical significance of *p* = 0.93. (**c**) Scatter plot showcasing the correlation between the full length of genes that encode ion channels and their A+T content. The Pearson correlation coefficient for this relationship is r = 0.32, with a level of statistical significance of *p* = 0.0003. (**d**) The five primary candidates (*GABRG1, GLRA3, TMEM38B, PKD2L2,* and *GLRB*) that have been sorted as the genes with relatively high mutability (the ‘Both’ group; Table d1). Five alternative candidate genes (*CLCN6, CLCNKA, CATSPER4, BEST4*, and *CHRNA9*), which have been sorted as the genes with relatively low mutability (the ‘None’ group; Table d2). (**e**) Scatter plot showcasing the proximity to telomeres of genes encoding 118 ion channels with respect to three sub-groups, which were determined by the full-length size of the gene (unit: base or b). The dotted horizontal line demarcates 50 Mb. After one-way ANOVA, no significant differences (*p* = 0.77) were found. (**f**) Scatter plot demonstrating the A+T content of genes encoding 118 ion channels with respect to the full-length of each gene. The dotted horizontal line demarcates 59%. Significant statistical differences were determined between the shortest (1 to 3000 bp) and longest size group (6001 to 17,000 bp) at *p* = 0.0002, as well as the intermediate (3001 to 6000 b) vs. the longest size group (6001 to 17,000 b) at *p* = 0.0096, respectively. **, *p* < 0.01; ***, *p* < 0.005.

**Figure 3 biomedicines-10-00594-f003:**
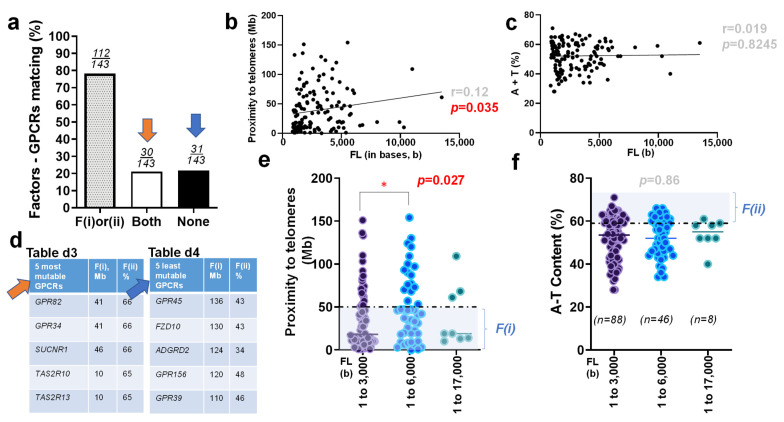
Relative mutability of druggable GPCRs by two factors (**a**) Bar graph, which summarizes the matching rate of genes that encode 143 druggable GPCRs with their F(i), proximity to telomeres and F(ii), their A+T content. Genes are divided into three groups: those that meet one of the two factors, (F(i) or (ii)), those that satisfy both F(i) and F(ii) (Both, denoted with orange arrow), and those that meet neither F(i) nor F(ii) (None, denoted with blue arrow) (**b**) Scatter plot that exhibits the relationship between the full-length size of genes that encode GPCRs and their associated proximity to telomeres. Statistical analysis resulted in a Pearson correlation coefficient of r = 0.12, with a level of statistical significance of *p* = 0.035. (**c**) Scatter plot that displays the relationship between the full length of genes that encode GPCRs and their A+T content. The Pearson correlation coefficient was determined to be r = 0.019, while the level of statistical significance was *p* = 0.8245. (**d**) Five primary candidates (*GPR82, GPR34, SUCNR1, TAS2R10*, and *TAS2R13*) that have been sorted as genes with relatively high mutability (the ‘Both’ group; Table d3). Moreover, five other candidate genes (*GPR45, FZD10, ADGRD2, GPR156*, and *GPR39*) have been sorted as genes with relatively low mutability (the ‘None’ group; Table d4). (**e**) Scatter plot showcasing the proximity to telomeres of genes encoding 143 GPCRs, which have been divided into three sub-groups by the full-length size of their gene (bp). A dotted horizontal line demarcates 50 Mb. A statistically significant difference between the smallest (1 to 3000 bp) and intermediate size groups (3001 to 6000 b) was determined as *p* = 0.027 (**f**) Scatter plot indicating the relationship between the A+T content of genes encoding 143 GPCRs with respect to the full-length of each of the genes. A dotted horizontal line demarcates 59%. No significant difference was found among any of these groups. * *p* < 0.05.

**Figure 4 biomedicines-10-00594-f004:**
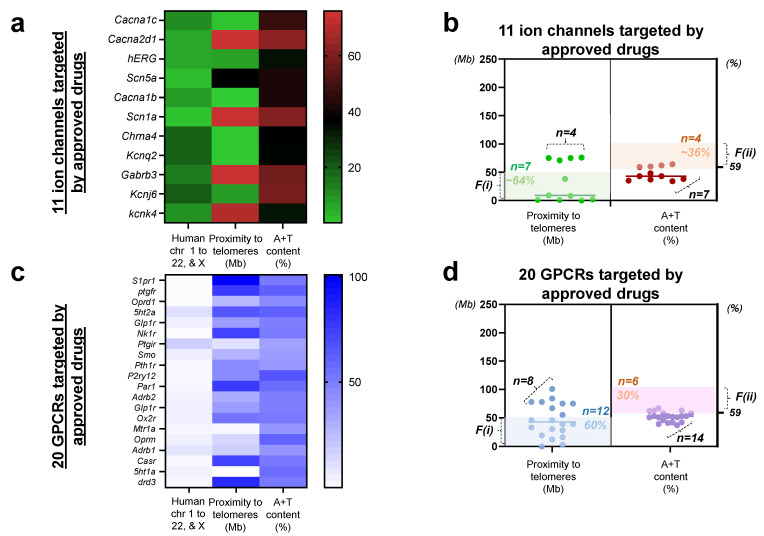
Relative mutability of ion channels and GPCRs targeted by approved drugs (**a**) the color-coded map illustrating distributions of 11 genes encoding ion channels targeted by commercialized drugs with respect to the human chromosome 1 through 22 and X (color code range 1 to 23), proximity to telomeres (range 0 to 100), and A+T content (range 0 to 100), respectively. Note that all 11 genes are marked on the left (y) axis. The reference number with coded color is shown on the right (**b**) scatter plots displaying relative mutability of 11 ion channels by proximity to telomeres (**left**) and A+T content (**right**). Shaded regions with light green (**left**) and light brown (**right**) demonstrate genes satisfying either of the two factors. Note that genes outside the shaded regions (*n* = 4 by F(i); *n* = 7 by F(ii)) are relatively less mutable. (**c**) the color-coded map illustrating distributions of 20 genes encoding GPCRs with respect to the human chromosome 1 through 22 and X (color code range 1 to 23), proximity to telomeres (range 0 to 100), and A+T content (range 0 to 100), respectively. Note that all 20 genes are marked on the left (y) axis. The reference number with coded color is shown on the right (**d**) scatter plots exhibiting relative mutability of 20 GPCRs by proximity to telomeres (**left**) and A+T content (**right**). Shaded regions with light blue (**left**) and light brown (**right**) show genes satisfying either of the two factors. Note that genes outside the shaded regions (*n* = 7 by F(i); *n* = 15 by F(ii)) are relatively less mutable.

**Figure 5 biomedicines-10-00594-f005:**
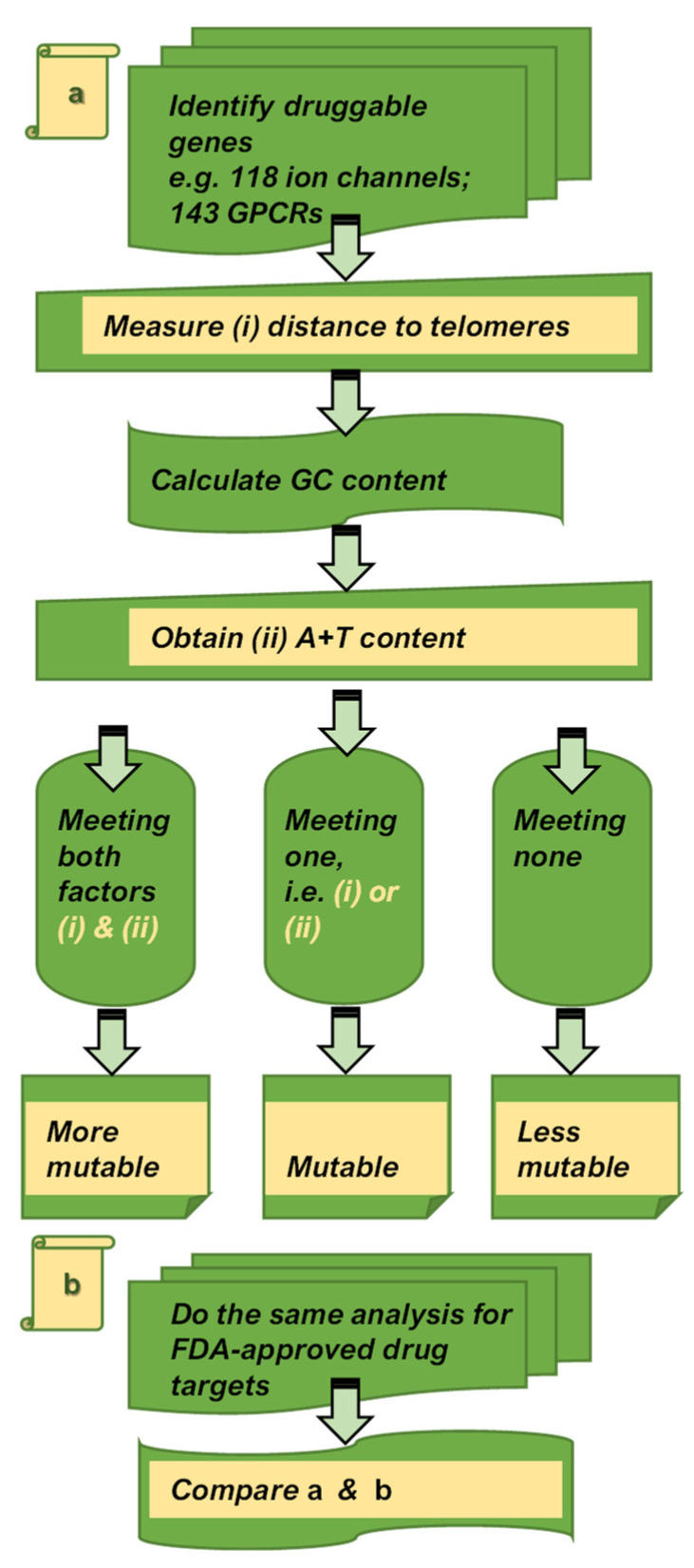
Flow chart summarizing how chromosomal analyses are conducted in this study (**a**) diagrams showing the logical flow of tasks on druggable genomes (**b**) diagrams displaying tasks on FDA-approved drug targets and amalgamation of the dataset in a and b.

**Table 1 biomedicines-10-00594-t001:** Two factor characteristics of 11 ion channels targeted by approved drugs.

Drug	Target Gene	Chr	Gene Loci	Telomere Loci	Proximity (Mb)	A,T (%)	A+T (%)	FL Size (bp)	References
Amlodipine	*CACNA1C*	12	1.9	0	1.9	23,25	48	13,744	[185]
Pregabalin	*CACNA2D1*	7	82	158	76	31,33	64	7542	[185]
Sotalol	*HERG (KCNH2)*	7	150	158	8	16,19	35	4292	[185]
Flecainide	*SCN5A*	3	38	0	38	21,22	43	8516	[185]
Ziconotide	*CACNA1B*	9	138	138.1	0.1	21,22	43	9792	[185]
Lacosamide	*SCN1A*	2	166	241	75	30,32	62	13,079	[185]
Varenicline	*CHRNA4*	20	63	64	1	17,21	38	5583	[185]
Retigabine	*KCNQ2*	20	63	64	1	17,20	37	9163	[185]
Diazepam	*GABRB3*	15	26	101	75	28,31	59	5767	[185]
VU0456810	*KCNJ6*	21	37	46	9	29,31	60	19,659	[185,186,187]
Riluzole	*KCNK4*	11	64	135	71	14,20	34	1829	[188]

*** marked in grey indicating the relatively less mutable characteristics, meeting none of the two factors.

**Table 2 biomedicines-10-00594-t002:** Two factor characteristics of 20 GPCRs targeted by approved drugs.

Drug	Target Gene	Chr	Gene Loci	Telomere Loci	Proximity (Mb)	A,T (%)	A+T (%)	FL Size (bp)	References
Siponimod	*S1PR11*	11	101	0	101	24,29	53	2778	[189]
Latanoprostene bunod	*PTGFR*	1	78	0	78	30,33	63	5429	[189]
Hycodan	*OPRD* or *OPRD1*	1	28	0	28	21,25	46	9317	[60]
Rexulti	*5HT2A* or HTR2A	13	46	113	67	31,32	63	5415	[60]
Trulicity	*GLP1R*	6	39	0	39	24,27	51	6682	[60]
Varubi	*NK1R* or *TACR1*	2	75	0	75	25,27	52	4779	[60]
Uptravi	*PI2R* or *PTGIR*	19	46	58	12	17,20	37	2078	[60]
Odomzo	*SMO*	7	129	159	30	17,22	39	3977	[60]
Tymlos	*PTHR1* or *PTH1R*	3	46	0	46	20,21	41	2153	[60]
Kengreal	*P2Y12* or *P2RY12*	3	151	197	46	34,33	67	2244	[60]
Zontivity	*PAR1* or *NR1I2*	3	197	197	0.1	33,25	58	2232	[60]
Striverdi Respimat	*ADRB2*	5	148	181	33	23,28	51	2013	[60]
Adlyxin	*GLP1R*	6	39	0	39	24,27	51	6682	[60]
Belsomra	*OX2R* or *HCRTR2*	6	55	0	55	24,29	53	1952	[60]
Hetlioz	*MTR1A* or *MTNR1A*	4	186	189	3	19,23	42	1289	[60]
Symproic	*OPRM* or *OPRM1*	6	154	170	16	30,32	62	15,143	[60]
Northera	*ADRB1-3* or *ADRB1*	10	114	133	19	18,24	42	3039	[60]
Parsabiv	*CASR*	3	122	197	75	26,27	53	10,062	[60]
Aristada	*5HT1A* or *HTR1A*	5	197	197	0.1	33,25	58	2232	[60]
Vraylar	*DRD3*	3	114	198	84	25,28	53	2770	[60]

*** marked in grey indicating the relatively less mutable characteristics, meeting none of the two factors.

## Data Availability

The data presented in this study are available in Appendix A.

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
