# Peer review of "Drug-Targeted Genomes: Mutability of Ion Channels and GPCRs"

_biomedicines, 2022, doi:10.3390/biomedicines10030594_

Round 1

Reviewer 1 Report

This review by Raines et al., provides an exciting new approach for the mapping of druggable genes and potential drug targets. The authors use two factors to predict mutability, a measure of proximity to telomeres, and adenine + thymine content. By using 390 understudied druggable genomes (in the publicly open NIH program announcement-PA-19-034), the authors examined the mutability of 118 ion channel and 143 G protein-coupled receptors (GPCRs). The authors found that 68% of their ion channel target genes, and 78% of their GPCR target genes were highly mutable consistent with two independent factors. The author also found that 20 GPCRs targeted by FDA-approved drugs have a relatively lower mutability than the 11 ion channels targeted by FDA-approved drugs, suggesting that the matching factor may be a major determinant for selecting the mutable targets. Their study also found that the mutability of ion channels and GPCRs were associated with their relative sizes, suggesting that findings from this study could become a basic principle for targeted drug development.   Overall, the author's work provides a much richer understanding of the potential druggable targets by identifying factors such as telomere proximity and nucleotide adenine + thymine content that has predictive power in predicting mutability, thereby allowing more efficient and predictive drug discovery. This article is certainly a great advancement in the field, and it opens new opportunities for understanding the genes, pathways, and mechanisms involved in disease. The author did a good job explaining the current state of knowledge in this field. The articles are very well written and easy to understand.

Here are very few comments that the author needs to consider:

  1. A summary figure describing the layout of the method carried out in the paper would be helpful. A general flow diagram of your analysis method will be especially distance to telomeres and adenine/thymine content on ion channels, and GPCRs would be very helpful for a wide range of readers.
  2. Mutability of ion channels: Several ion channels such as sodium, potassium, calcium, cation, or anion channels that are altered in diseases. In this study 118, ion channels genes were examined, so which channel type was predominant in this pool. Is there any correlation of any specific channel mutability based on these factors telomere proximity and nucleotide adenine + thymine content? It would be to show which particular channel type (for example, potassium or sodium or cation or anion) is more likely to have a high mutability based on your analysis?
  3. Section 3.5 and Table 1: Why the ion channel gene name is in the small letters? line 327, cacna1c, line 329, kcnh2 (in the Table, it is KCNH2)....
  4. which one is correct?

Abstract, line 24: We also found that the GPCR genes (n=20) targeted by FDA-approved or, Conclusion, line 483:  twenty-one GPCRs targeted by FDA-approved drugs..

Aside from these comments, the article is very good, and I would recommend it to anyone who works in this field.

Author Response

Here are very few comments that the author needs to consider:

  1. A summary figure describing the layout of the method carried out in the paper would be helpful. A general flow diagram of your analysis method will be especially distance to telomeres and adenine/thymine content on ion channels, and GPCRs would be very helpful for a wide range of readers.

Thanks. Following your suggestion, we have added Fig. 5 Flow chart summarizing how chromosomal analyses are conducted in this study (a) diagrams showing the logical flow of tasks on druggable genomes (b) diagrams displaying tasks on FDA-approved drug targets and amalgamation of the dataset in a and b.

  1. Mutability of ion channels: Several ion channels such as sodium, potassium, calcium, cation, or anion channels that are altered in diseases. In this study 118 ion channels genes were examined, so which channel type was predominant in this pool. Is there any correlation of any specific channel mutability based on these factors telomere proximity and nucleotide adenine + thymine content? It would be to show which particular channel type (for example, potassium or sodium or cation or anion) is more likely to have a high mutability based on your analysis?

Thank you for your constructive question. To answer your question, at least in part, we added the new summary of the ion channel types in Supplementary Materials (Table S2). There are 30 cation channels identified while the majority (75% or 88 of 118 ion channels) belongs to cation channels. This list of understudied druggable ion channels (n=118) happens to pertain primarily to cation channels, therefore, the mutability result that we obtained is primarily obtained from cation channels.

Once we are able to establish almost the equal number of cation vs. anion channel, then, we will be able to establish an unbiased correlation between the ion channel type and the relative mutability by proximity to telomeres and/or A+T content. This should be a new study design for the next line of work.

  1. Section 3.5 and Table 1: Why the ion channel gene name is in the small letters? line 327, cacna1c, line 329, kcnh2 (in the Table, it is KCNH2)....

Thanks. To be consistent with Table S1-S3, we capitalized the gene abbreviations.

  1. which one is correct?

The source of novel druggable proteins was based on the NIH program announcement (https://grants.nih.gov/grants/guide/pa-files/PA-19-034.html). Here the druggable genomes are listed in capitalized abbreviation. Thus, we will follow this notation or “CAPITALIZED”.

Abstract, line 24: We also found that the GPCR genes (n=20) targeted by FDA-approved or, Conclusion, line 483:  twenty-one GPCRs targeted by FDA-approved drugs..

Thank you. n=20 is correct (abstract). Thus, we revised the conclusion accordingly from twenty-one to twenty.

Aside from these comments, the article is very good, and I would recommend it to anyone who works in this field. Thank you!

Reviewer 2 Report

In the article “Drug targeted genomes: Mutability of ion channels and GPCRs” by Raines et al., authors develop the factor-drugable gene matching rate to identify the mutability of druggable genes encoding ion channels and GPCRs. The article is valuable to the field, it is well written and comprehensive, concepts are clearly presented and discussed, giving information about the candidate genes and defining the associated characteristics with high mutation rates. The figures are appropriate and easy to read and understand.

I have only one suggestion to improve the final clarity of the article; given the relevance of this study in the clinical scenario, it could be interesting to analyze the future perspectives based on the data presented in this manuscript. This information could help the reader to perform a similar analysis with other pathological applications. Some grammatical mistakes can be easily corrected by careful revision of the English language employed. 

Author Response

In the article “Drug targeted genomes: Mutability of ion channels and GPCRs” by Raines et al., authors develop the factor-drugable gene matching rate to identify the mutability of druggable genes encoding ion channels and GPCRs. The article is valuable to the field, it is well written and comprehensive, concepts are clearly presented and discussed, giving information about the candidate genes and defining the associated characteristics with high mutation rates. The figures are appropriate and easy to read and understand.

I have only one suggestion to improve the final clarity of the article; given the relevance of this study in the clinical scenario, it could be interesting to analyze the future perspectives based on the data presented in this manuscript. This information could help the reader to perform a similar analysis with other pathological applications. Some grammatical mistakes can be easily corrected by careful revision of the English language employed. 

Response

We appreciate your comments. Following your suggestion, we have revised the conclusion of the study by adding the following paragraph:

”… Future perspectives: investigators in the field use laboratory animals aiming to test their hypotheses ultimately for translational purposes. It is not uncommon to find that a pharmacological approach that worked in rodents fails to reproduce the expected outcome when tested in human clinical trials. This study provides a guideline of prioritizing a more reliable drug target by relative mutability based on two factors using the genomes of humans as well as those of laboratory animals such as mice and rats. With all conditions such as safety, toxicity, and efficacy of drugs comparable and if there are two or more such drug targets, a less mutable pharmacological target can be determined, augmenting decision-making on which drug target will result in a consistent outcome in laboratory animals and humans. This means that the same analysis presented in this study can be applied to diverse animal genomes available at the NCBI database…”

Reviewer 3 Report

Authors present a paper on mutability of ion channels and GPCRs. I am not quite sure what was a main premise of the paper. I think that needs to be formulated better.

Major

  1. The section Conclusions need a serious rewrite to highlight the importance of the findings
  2. Abstract need similar improvement.
  3. Figure 1 in my opinion is redundant because Figure 2 conveys basically the same information.
  4. In Figure 3 and 4 panels b-c basically present the same data as panel e-f.

Minor

  1. P1 line 39: “selective” is not necessary. There are non-selective ion channels
  2. P2 line 76: broken reference. Please, also check formatting of other references because some references toward the end of the manuscript have different font.

Author Response

Comments and Suggestions for Authors

Authors present a paper on mutability of ion channels and GPCRs. I am not quite sure what was a main premise of the paper. I think that needs to be formulated better.

Major

  1. The section Conclusions need a serious rewrite to highlight the importance of the findings

We appreciate your comments. Following the suggestion, we have revised the conclusion of the study by adding the following paragraph:

”… Future perspectives: investigators in the field use laboratory animals aiming to test their hypotheses ultimately for translational purposes. It is not uncommon to find that a pharmacological approach that worked in rodents fails to reproduce the expected outcome when tested in human clinical trials. This study provides a guideline of prioritizing a more reliable drug target by relative mutability based on two factors using the genomes of humans as well as those of laboratory animals such as mice and rats. With all conditions such as safety, toxicity, and efficacy of drugs comparable and if there are two or more such drug targets, a less mutable pharmacological target can be determined, augmenting decision-making on which drug target will result in a consistent outcome in laboratory animals and humans. This means that the same analysis presented in this study can be applied to diverse animal genomes available at the NCBI database…”

  1. Abstract need similar improvement.

Thanks. We revised the abstract accordingly:

“…Mutations of ion channels and G protein-coupled receptors (GPCRs) are not uncommon and can lead to cardiovascular diseases. Given previously reported multiple factors associated with high mutation rates, we sorted the relative mutability of multiple human genes by (i) proximity to telomeres and/or (ii) high adenine and thymine (A+T) content. We extracted genomic information using the genome data viewer and examined the mutability of 118 ion channel and 143 GPCR genes based on their association with factors (i) and (ii). We then assessed these two factors with 31 genes encoding ion channels or GPCRs that are targeted by the United States Food and Drug Admin-istration (FDA)-approved drugs. Out of the 118 ion channel genes studied, 80 met either factor (i) or (ii), resulting in an 68% match. In contrast, a 78% match was found for the 143 GPCR genes. We also found that the GPCR genes (n=20) targeted by FDA-approved drugs have a relatively lower mu-tability than those genes encoding ion channels (n=11), where targeted genes encoding GPCRs were shorter in length. The result of this study suggests that the use of matching rate analysis on factor-druggable genome is feasible to systematically compare the relative mutability of GPCRs and ion channels. This chromosomal analysis by two factors identified a unique characteristic of GPCRs, which have a significant relationship between their nucleotide sizes and proximity to telomeres unlike most genetic loci susceptible to human diseases. Comparing these data with those of laboratory animals can provide a novel way of decision-making on which target gene might result in a consistent or less varied outcome over diverse animal genomes.

 …”

  1. Figure 1 in my opinion is redundant because Figure 2 conveys basically the same information.

Thanks. We removed Figure 2 and revised figure numbers in results, figure legend, and discussion accordingly.

  1. In Figure 3 and 4 panels b-c basically present the same data as panel e-f.

Thanks. In a way they look similar but if one conducts ANOVA of three groups, one will also conduct a pair-wise comparison after one-way ANOVA to compare a specific difference between the groups. Likewise, panels b-c of Figure 3 & 4 would show a bigger picture like “ANOVA”, whereas e-f of Figure 3 & 4 would pertain to a specific picture like “post-hoc test after one-way ANOVA”. Thus, we would prefer to show both woods (b-c) and trees (e-f) at the same time.

Minor

  1. P1 line 39: “selective” is not necessary. There are non-selective ion channels

Thank you. We removed it as suggested.

  1. P2 line 76: broken reference. Please, also check formatting of other references because some references toward the end of the manuscript have different font.

Including this and Page 13, line 487, we revised into the proper references as suggested.

Reviewer 4 Report

The manuscript biomedicines-1605330 by Regan Raines et al., entitled “Drug targeted genomes: Mutability of ion channels and GPCRs” is an interesting molecular work reporting an interesting and potentially useful method to match rate analysis on the factor-druggable genome to systematically compare the relative mutability.

The manuscript is written with a flow, smooth and understandable language. The introduction is complete and provides sufficient background.

The research is designed appropriately as the methods described too.

The results are presented clearly, and the legends are explicit in the description.

The conclusions are supported by the Results.

In my opinion, the manuscript could be accepted for publication in the Biomedicine journal in the present form.

Author Response

Thank you!

Round 2

Reviewer 1 Report

The authors have satisfactorily addressed most of my concerns.

Reviewer 3 Report

In the revised manuscript, authors considerably improved presentation of the methods, results, and conclusions.